# Genome-Wide Identification and Expression Analysis of Expansin Gene Family in the Storage Root Development of Diploid Wild Sweetpotato *Ipomoea trifida*

**DOI:** 10.3390/genes13061043

**Published:** 2022-06-10

**Authors:** Ming Li, Lianfu Chen, Tao Lang, Huijuan Qu, Cong Zhang, Junyan Feng, Zhigang Pu, Meifang Peng, Honghui Lin

**Affiliations:** 1Key Laboratory of Bio-Resource and Eco-Environment of Ministry of Education, College of Life Sciences, Sichuan University, Chengdu 610065, China; lmww1981@sina.com; 2Biotechnology and Nuclear Technology Research Institute, Sichuan Academy of Agricultural Sciences, Chengdu 610061, China; langtao123xxx@126.com (T.L.); qhjuan120@126.com (H.Q.); zhangc1267@163.com (C.Z.); junyanfeng@live.cn (J.F.); zhigangpu@126.com (Z.P.); 3College of Plant Science and Technology, Huazhong Agricultural University, Wuhan 430070, China; chenlianfu@mail.hzau.edu.cn

**Keywords:** *Ipomoea trifida*, expansin, evolution, gene expression, storage root development

## Abstract

Expansins play important roles in root growth and development, but investigation of the expansin gene family has not yet been reported in *Ipomoea trifida*, and little is known regarding storage root (SR) development. In this work, we identified a total of 37 *expansins* (*ItrEXPs*) in our previously reported SR-forming *I. trifida* strain Y22 genome, which included 23 *ItrEXPAs*, 4 *ItrEXPBs*, 2 *ItrEXLAs* and 8 *ItrEXLBs*. The phylogenetic relationship, genome localization, subcellular localization, gene and protein structure, promoter *cis*-regulating elements, and protein interaction network were systematically analyzed to reveal the possible roles of *ItrEXPs* in the SR development of *I. trifida*. The gene expression profiling in Y22 SR development revealed that *ItrEXPAs* and *ItrEXLBs* were down-regulated, and *ItrEXPBs* were up-regulated while *ItrEXLAs* were not obviously changed during the critical period of SR expansion, and might be beneficial to SR development. Combining the tissue-specific expression in young SR transverse sections of Y22 and sweetpotato tissue, we deduced that *ItrEXLB05*, *ItrEXLB07* and *ItrEXLB08* might be the key genes for initial SR formation and enlargement, and *ItrEXLA02* might be the key gene for root growth and development. This work provides new insights into the functions of the expansin gene family members in *I. trifida*, especially for *EXLA* and *EXLB* subfamilies genes in SR development.

## 1. Introduction

Plant cell walls consist of cellulose microfibrils and matrix polysaccharides, and cell walls serve as protective barriers, and provide structure supporting the size and shape of the cells [1]. Expansins are cell wall-loosening proteins with non-enzymatic activity that participate in the regulation of cell wall extension and relaxation by inducing the slippage of cellulose microfibrils, thus promoting cell enlargement or development [2]. Plant expansins are typically 250–275 amino acids in length and are composed of DPBB_1 and Pollen_allerg_1 domains preceded by a signal peptide (SP); on the basis of the phylogenetic relationship of protein sequences, plant expansins can be divided into α-expansin (EXPA), β-expansin (EXPB), expansin-like A (EXLA) and expansin-like B (EXLB) subfamilies [3]. EXPA and EXPB have been demonstrated to cause cell wall loosening, whereas little is known about EXLA and EXLB [4].

Expansin genes have been identified in many plants, such as *Arabidopsis thaliana*, rice, maize, and wheat [3,5] and many studies have focused on root elongation, root hair initiation and lateral root formation [6,7,8,9,10]. Sweetpotato (*Ipomoea batatas*), the seventh most important food crop species worldwide, is harvested from its below-ground storage root (SR) [11]. Sweetpotato is a hexaploid (B_1_B_1_B_2_B_2_B_2_B_2_) with 90 chromosomes, which has a high degree of heterozygosity and high number of repetitive sequences [12,13], thus hindering gene identification and functional research, especially those involved in SR development. To date, only several genes have been reported involved in sweetpotato SR development, such as *IbEXP1*, *IbMADS1*, *SRD1*, *SRF1*, *KNOXI* and *IbBBX24* [14,15,16,17,18]. Among them, *IbEXP1* is the first expansin gene that has been experimentally shown to be involved in sweetpotato SR development [14]; and other two expansin genes, *IbEXP2* and *IbEXPL1*, have also been reported to be related to sweetpotato SR development at the transcriptional regulation level [15,16]. However, the function of other genes in the expansin gene family is still unknown due to the limitation of sweetpotato genome research. *I. trifida* is the putative progenitor of sweetpotato, and is a species complex with diploids to hexaploids [19,20,21]. The diploid *I. trifida* has a relatively simple genome and has gradually become a model for sweetpotato research, especially with respect to SR formation [22,23,24,25,26]. Therefore, research on expansin genes and their roles in SR development of *I. trifida* could be helpful for an improved understanding of the mechanisms underlying SR development in both *I. trifida* and sweetpotato.

In this work, the diploid *I. trifida* strain Y22 [25], which has typical SR-forming characteristics, was used as material. The possible roles of the expansin gene family in SR development were systematically identified and predicted. Combining different analyses, we deduced that the expansin gene family might play an important role in initial SR formation and SR swelling, *ItrEXLB05*, *ItrEXLB07* and *ItrEXLB08* might be the key genes for initial SR formation and enlargement, and *ItrEXLA02* might be the key gene for root growth and development. Our work provides evidence for further studying the function of specific expansin genes in *I. trifida* and could be helpful for further using in genetic improvement of sweetpotato.

## 2. Materials and Methods

### 2.1. Identification of Expansin Genes in I. trifida

Y22 is a diploid *I. trifida* strain and has good SR-forming characteristics [25]. The genome and annotation file (GFF3 format) of Y22 were provided by our research group [25]. The expansin protein sequences of *Arabidopsis thaliana* were obtained from TAIR (https://www.arabidopsis.org/ (accessed on 4 December 2021)). Firstly, the longest protein sequence of each gene was obtained by removing any alternatively spliced sequences based on the genome annotation file (with a Perl script). Secondly, the whole genome protein sequences of *I. trifida* were scanned as query sequences with BLASTP (evalue < 1 × 10^−^^5^) [27] against the standard expansin protein sequences of *A. thaliana*, and then the homologous expansin sequences were obtained. Thirdly, these protein sequences were validated with the two protein families HMM models DPBB_1 (PF03330) and Pollen_allerg_1 (PF01357) in Pfam database (33.1) [28] by HMMER (3.3.1) [29] (evalue < 1 × 10^−5^ and coverage of HMM model ≥ 0.25) method, and those protein sequences containing both DPBB_1 and Pollen_allerg_1 domains were retained. Finally, the candidate sequences were further confirmed using SignalP-5.0 (https://services.healthtech.dtu.dk/service.php?SignalP-5.0 (accessed on 17 December 2021)), and the proteins containing an SP were considered as ItrEXPs. Then, the chemical properties and subcellular localization of all expansins were predicted by ExPASy [30] and Plant-mPLoc (2.0) [31], respectively.

### 2.2. Sequence Alignment and Phylogenetic Analysis

The expansin proteins from Y22, *A. thaliana*, *Ipomoea nil* [32] and *Ipomoea triloba* [24] were subsequently included in the phylogenetic analysis. Multiple sequence alignment and neighbor-joining (NJ) phylogenetic tree were performed by the MEGA X program with default parameters. The phylogenetic tree was then visualized by the Evolview v3 [33].

### 2.3. Chromosomal Localization and Gene Duplication Analysis

The chromosomal positions and collinearity of all the *ItrEXPs* were analyzed from the GFF3 file. Synteny analysis within the Y22 genome was conducted by MCscan (Python version) [34], after which TBtools (v1.098685) [35] was used to display the segmentally duplicated *ItrEXPs*. To analyze the duplication events of all the *ItrEXPs* resulting from a WGT event [24,25], the *Coffea canephora* genome sequence information [36] was downloaded from the NCBI database (https://www.ncbi.nlm.nih.gov/ (accessed on 4 December 2021)). The microsynteny between the *C. canephora* and Y22 genomes was analyzed and visualized by MCscan (Python version) [34].

### 2.4. Conserved Motif, Gene Structure, and Promoter Analysis

The motifs were predicted by the MEME (5.4.1) (http://meme-suite.org/tools/meme (accessed on 8 January 2022)). To gain insights into the gene structure, the GFF3 information of all the *ItrEXPs* was extracted by a Perl script, and then GSDS (v2.0) [37] was used to generate gene structure graphs. The potential cis-elements in the promoter region was predicted by plantCARE [38], then visualized by TBtools (v1.098685) [35].

### 2.5. Prediction of Protein Interaction

The protein interaction networks of expansins were predicted by STRING [39] based on *A. thaliana* homologous proteins, and the network map was modified by Adobe Illustrator CS6.

### 2.6. RNA-Sequencing (RNA-Seq) and Analysis

The RNA-seq datas from samples taken at four typical stages of Y22 SR development, including the adventitious root (AR, S0), initial storage root (ISR, S1, diameter ≤ 2 mm), young storage root (YSR, S2, 5–8 mm), and mature storage root (MSR, S3, ≥20 mm) stages, according to the reference [25]. Total RNA was extracted using Invitrogen^TM^ TRIzol^TM^ reagent (Life Technologies Co., Carlsbad, CA, USA) and treated with RNase-free DNase I (Promega Co., Madison, WI, USA), respectively. The RNA concentration and purity were measured using a NanoDrop 2000 (Thermo Fisher Scientific, Co., Ltd., Waltham, MA, USA). RNA integrity was assessed using RNA Nano 6000 Assay Kit of Agilent Bioanalyzer 2100 system (Agilent Technologies Co., Ltd., Palo Alto, CA, USA). A total amount of 1 μg RNA per sample was used as input material for sequencing library. The libraries were generated using the NEBNext Ultra RNA Library Prep Kit (New England BioLabs, Ltd., Ipswich, MA, USA) and sequenced by the HiSeq Xten (Illumina Inc., San Diego, CA, USA). The RNA-seq raw reads was filtered and trimmed by Trimmomatic [40], the RNA-seq clean reads mapping to Y22 genome was conducted by TopHat2 [41] with the same parameters as the reference [25]. The RPKM (reads per kilobase per million mapped reads) value and read count were calculated by StringTie2 [42] and HTSeq [43], respectively. Expression count matrix files were used to analyze expression differences via DESeq2 [44] (*p* < 0.05).

### 2.7. Quantitative Real-Time PCR (qRT-PCR) Analysis

Total RNA isolated for RNA-seq was used for qRT-PCR analysis the expression of random selected expansin genes in four typical stages of Y22 SR development. The YSR from Y22 and sweetpotato cv. Nancy Hall were sliced transversely into five sections as in the reference [25], for analysis the expression of *ItrEXPs* in the different sites of growing SR. Total RNA was isolated as described above for RNA-Seq, and reverse transcribed using a PrimeScript™ RT reagent Kit with gDNA Eraser (TaKaRa Biomedical Technology (Dalian) Co., Ltd., Dalian, China) in accordance with the manufacturer’s instructions. qRT-PCR was performed using a ChamQ Universal SYBR qPCR Master Mix (Vazyme Biotech Co., Ltd., Nanjing, China) in a Roche LightCycler 96 (Roche Diagnostics, Mannheim, Germany) according to the manufacturer’s instructions. The PCR conditions were 5 min at 95 °C, followed by 40 cycles of 10 s at 95 °C, 10 s at 58 °C and 10 s at 72 °C. *I. trifida Actin* gene was used as an internal control [25]. The primer pairs of selected genes are listed in Appendix A. The 2(-Delta Delta C(T)) method was used for calculating the gene relative expression [45].

### 2.8. Gene Expression Analysis of Development Responsive Interacted Proteins

The sequences of development related interaction proteins were extracted from *A. thaliana* based on their protein IDs and used as standard sequences. The homologous of the interaction proteins in Y22 genome were obtained by BLASTP (evalue < 1 × 10^−^^5^) [27] against these standard protein sequences of *A. thaliana*. The gene expression was calculated using the results of RNA-seq analysis by the homologous gene ID.

## 3. Results

### 3.1. Identification and Characterization of Expansin Family Members in Y22

Based on 36 *A. thaliana* expansin proteins from TAIR, a total of 37 ItrEXPs were identified in Y22 genome, and all these ItrEXPs contain both the conserved domains of the DPBB_1 (PF03330) and Pollen_allerg_1 (PF01357) and expansin SP. The 37 ItrEXPs were classified into four subfamilies (ItrEXPA, ItrEXPB, ItrEXLA and ItrEXLB), of which there were 23, 4, 2 and 8 members, respectively (Table 1). Further analysis revealed that the number of amino acid (AA) residues varied between 238 and 310 with an average of 258; the average molecular weight (MW) was 27.81 kD, ranging from 25.49 to 32.70 kD; and the isoelectric point (pI) ranged from 4.62 to 10.42, with 14 proteins being acidic and 23 being alkaline (Table 1). The subcellular localization results suggested that all the ItrEXPs were located on the cell wall, indicating that their function is related to cell growth.

### 3.2. Phylogenetic Analysis of Expansins

To analyze the evolutionary relationships of expansin proteins between *I. trifida* and its wild relatives, the homologous proteins from Y22, *I*. *nil* [32] (Appendix A), *I*. *triloba* [24] (Appendix A) and *A. thaliana* were used to construct an unrooted NJ phylogenetic tree (Figure 1). According to the phylogenetic tree, the expansins could be divided into four clades, which correspond to the EXPA, EXPB, EXLA and EXLB subfamilies, respectively. Compared with *I. triloba*, Y22 has less members in EXPA, EXPB and EXLB subfamilies, and has similar members in EXLA. Compared with *I. nil,* Y22 has one more member in EXLA and EXLB subfamilies, respectively. In the phylogenetic tree, expansin members of Y22, *I. triloba* and *I. nil* in subclass I and III of EXPA subfamily, and in EXPB and EXLA families, were together in a small clade, respectively, which indicated that the neighboring expansin proteins of *I. trifida* (Y22), *I. nil* and *I. triloba* have a more closely phylogenetical relationship (Figure 1).

### 3.3. Chromosomal Location and Gene Duplication

The genomic location results showed that the *ItrEXPs* were distributed unevenly on the chromosomes of Y22, and the numbers of every chromosome was quite different. The chromosome 15 (Chr15) with nine *ItrEXPs* had the highest density, but Chr10 had no *ItrEXPs* (Figure 2a). To further reveal the duplication type of the *ItrEXPs*, synteny analysis within the Y22 genome was conducted by MCscan [34]. The results showed that nine tandem duplication genes were found to form four gene clusters (*ItrEXPA05*/*ItrEXPA06* on Chr02, *ItrEXPA07*/*ItrEXPA08* on Chr04, *ItrEXLB02*/*ItrEXLB03* on Chr08, *ItrEXPA19*/*ItrEXPA20*/*ItrEXPA21* on Chr15), and seven segmental duplication events with fourteen *ItrEXPs* were discovered (Figure 2a, Appendix A). These results indicated that not only tandem duplication but also segmental duplication occurred throughout the evolution of *ItrEXPs* gene family.

To further explore the phylogenetic relationship of expansin genes resulting from the WGT event [24,25], we compared the genome collinearity between Y22 and *C. canephora*. Seventeen expansin genes of *C. canephora* exhibited a syntenic relationship with 18 *ItrEXPs* of Y22 (Appendix A). Among them, there were one-to-one, one-to-two (i.e., *gene-GSCOC_T00016583001* to *ItrEXPA13* and *ItrEXPA17*), and one-to-three (i.e., *gene-GSCOC_T00028253001* to *ItrEXPA12*, *ItrEXPA10* and *ItrEXPA15*) syntenic genes between *C. canephora* and Y22 (Figure 2b–d), indicating that there are some expansin family members of the *I. trifida* (Y22) that underwent duplication or triplication in the WGT event throughout evolutionary history. Besides, there were two-to-one (i.e., *gene-GSCOC_T00000261001* and *gene-GSCOC_T00037639001* to *ItrEXLB08*) and four-to-one (i.e., *gene-GSCOC_T00027140001*, *gene-GSCOC_T00027139001*, *gene-GSCOC_T00027142001* and *gene-GSCOC_T00027143001* to *ItrEXLB03*) syntenic genes between *C. canephora* and Y22 (Figure 2e,f), indicating that gene loss may occur in the *ItrEXP* gene family within or after the WGT event.

### 3.4. Protein Domains and Gene Structure Analysis

The analysis of motif diversity across all ItrEXPs in Y22 showed that a total of 15 different motifs were identified (Appendix A); the protein structure of each subfamily was relatively conserved, but still had some differences (Figure 3a,b). Such as the ItrEXPA subfamily, the compositon of motifs 1, 2, 3, 6 and 7 were conserved among the members; however, the motifs 5, 8, 9, 10, 14 and 15 of ItrEXPA16, ItrEXPA18, ItrEXPA19, ItrEXPA20 and ItrEXPA21 were different. In the ItrEXPB subfamily, compared with ItrEXPB01, ItrEXPB03 and ItrEXPB04, ItrEXPB02 lacked motifs 7 and 12. All the results indicated that the ItrEXPs with the same motifs may perform similar functions in *I. trifida* (Y22).

Furtherly, the intron/exon structural diversity of the *ItrEXPs*, was performed. The results showed that the intron numbers of *ItrEXPs* ranged from zero to four (Figure 3c). Seventeen of 23 *ItrEXPA* members had 2 introns, *ItrEXPA12* and *ItrEXPA23* had 1 intron, but *ItrEXPA21* had no introns; however, most of the *ItrEXPB*, *ItrEXLA* and *ItrEXLB* genes had 3 introns except *ItrEXLA02,* and *ItrEXLB08* had 4 introns, suggesting that there is a tight evolutionary relationship between members in the same subfamily. These differences in intron loss or gain may be due to the loss or acquisition of introns throughout long-term evolution.

### 3.5. Potential Cis-Element Analysis in Gene Promoter Regions

To further reveal the function of *ItrEXPs*, the *cis*-elements in promoter regions were predicted by plantCARE [38]. The results showed that there were many types of *cis*-elements, including core elements (CAAT-box and TATA-box) and binding sites, development-related, hormone-related, light-related, and stress/defense-related elements. Eleven development-responsive elements (A-box, TCA-element, RY-element, CAT-box, GCN4_motif, O2-site, HD-Zip 1, CCAAT-box, circadian, SARE and MBSI) were found in most of the *ItrEXPs* (Figure 4), indicating that most of the *ItrEXPs* might play a crucial role in development. In addition, 8 hormone-responsive *cis*-elements (the auxin, JA, GA and ABA related elements) (Figure 4), 24 light and 6 stress/defense responsive cis-elements (Appendix A) were found. A previous study has shown that hormone, light and stress/defense response could change the acid environments of cell wall, while expansins are ‘acid-induced growth’ proteins [2]. All the results indicated that *ItrEXPs* might be involved in the crosstalk between different hormone and environment signaling pathways to change the acid environments of cell walls, thus regulating growth and development.

### 3.6. Protein Interaction Network of Expansins

To explore the potential regulatory network of ItrEXPs in Y22, the protein interaction networks were predicted by STRING [39]. The results showed that ItrEXPA07 might interact with three ItrEXPAs, and ItrEXPA22 with five ItrEXPAs; notably, ItrEXLA01 might interact with ItrEXLA02 and ItrEXLB08 (Figure 5a), suggesting that protein–protein interactions occur not only within subfamilies, but also between subfamilies. We also found that ItrEXPs might interact with some development-related proteins (i.e., HBI1, ATPMEPCRC, CEV1, XTHs, XTR6, TCH4, EXO, EXL4, MGD2, VGD1, KIN13A). Among them, HBI1 may act as a positive regulator of cell elongation downstream of multiple external and endogenous signals by direct binding to the promoters and activation of the *ItrEXPA21* [39]; XTHs, XTR6 and TCH4 may interact with ItrEXPAs, ItrEXPBs and ItrEXLAs, which could cleave and religate xyloglucan polymers and participate in cell wall construction (Figure 5b) [39]; KIN13A may interact with ItrEXLBs, which could participate in cell wall construction of metaxylem vessel cells [39]. In addition, expansin proteins might interact with the pectin lyase-like superfamily proteins (i.e., AT3G09540, AT3G27400, AT4G24780, PME5 and QRT3) and regulating factors (i.e., GRF1, GRF5, AN3 and RALF1) (Figure 5b). These results indicated that expansin proteins may participate in the extension and reconstruction of cell wall and promote cell enlargement. It was noted that ten ItrEXPAs might interact with the auxin-responsive proteins (AT2G21200, SAUR15, SAUR19 and IAA19) and GA-regulated proteins (GASA5 and GASA6) (Figure 5c), which could change the acid environments of cell wall. Taken together, the results of protein interaction networks revealed that ItrEXPs might involved in the extension and reconstruction of cell wall and participate in hormone signaling pathways and play an important role in growth and development.

### 3.7. Expression Analysis of ItrEXPs in Y22 SR Development

To determine the probable roles of *ItrEXPs* in Y22 SR development, roots at four typical stages, including the AR, ISR, YSR and MSR stages, were sampled for RNA-seq (Figure 6a). The results showed that 22 members out of the 37 *ItrEXPs* were expressed (RPKM > 1) during Y22 SR development (Figure 6b). Among the 12 expressed *ItrEXPAs*, six genes were down-regulated, three genes were up-regulated slightly; two of the three *ItrEXPBs* were up-regulated; and four of the six *ItrEXLBs* were down-regulated. The results showed that the down-regulated expression of *ItrEXPA* and *ItrEXLB* subfamily genes might be required during SR development, indicating that *ItrEXPAs* and *ItrEXLBs* might have similar function, which are similar to the reported low expression of *IbEXP1* to promote SR enlargement [14]. While *ItrEXPBs* might have the opposite function from *ItrEXPAs* and *ItrEXLBs*, requiring up-regulated expression during SR development. Notably, the expression levels of *ItrEXLB05*, *ItrEXLB07* and *ItrEXLB08* in the AR stage were more than ten times higher than those in the key stages of ISR, suggesting that these three genes might play a key regulatory role in SR formation (Figure 6b). Interestingly, the expression level of the *ItrEXLA02* did not change significantly during the entire root development process, and had been maintained at a high level, indicating that the maintenance of its high expression may also play an important role in the formation, growth, and development of the root (Figure 6b).

To validate the RNA-Seq results, 13 members randomly selected from the expressed *ItrEXPs* were further analyzed by qRT-PCR analysis. The results showed that the gene expression patterns were consistent with the results of RNA-seq (Figure 6b,c).

### 3.8. Expression Analysis of ItrEXPs in Different Tissues of SR inY22 and Sweetpotato

To determine the probable roles of predicted key *ItrEXPs* in the enlargement of SR, the YSR from Y22 and sweetpotato cv. Nancy Hall (NH) were sliced transversely and divided into five sections (Figure 7a). Several representative genes from each subfamily were selected for gene expression analysis in different sections of SR. On the whole, the expression levels of *ItrEXPAs*, *ItrEXPBs* and *ItrEXLBs* were higher, at least 10-fold, in Y22 than those in NH, while the *ItrEXLA02* in Y22 was similar to NH (Figure 7b–i); in spite of this, these genes exhibited similar expression patterns in both Y22 and sweetpotato, suggesting that relatively low expression of *ItrEXPAs*, *ItrEXPBs* and *ItrEXLBs* might be more beneficial to SR growth and expansion, while *ItrEXLA02* had little correlation with SR expansion and might play a role in maintaining SR growth. Notably, the expression levels of *ItrEXPAs* and *ItrEXPBs* were higher in Section 2 (SC2) than those in other parts, while *ItrEXLA02* and *ItrEXLBs* showed higher expression in Section 3 (SC3) to Section 5 (SC5) (Figure 7b–i). The results indicated that *ItrEXPs* showed tissue-specific expression in YSR, and the main roles of *ItrEXPAs* and *ItrEXPBs* might be in the cambium, while the main roles of *ItrEXLA02* and *ItrEXLBs* might be in the stele.

## 4. Discussion

Among the 600–700 species of the genus *Ipomoea*, at least 63 have been recorded as having SR, several of which are edible and some of which are larger than those of sweetpotato; however, *I. trifida* is not among these 63 species [21]. On the basis of a phylogenetic analysis together with morphological studies, DNA barcodes and high-throughput sequencing, *I. trifida* has been determined to be the closest wild relative of sweetpotato, and sweetpotato may have diverged from *I. trifida* more than one million years ago [19,20,21]. Compared with sweetpotato, the diploid *I. trifida* has much smaller genome size and chromosome number, and the genome structure is also simpler. Therefore, the diploid *I. trifida* has been used as a model plant species for gene exploring and identification in sweetpotato SR development [22,23,24,25,26]. However, diploid *I. trifida* reported before had no SR, only one report about SR-like *I. trifida* [46], which seriously limited the study on the development of SR in *I. trifida*. The diploid *I. trifida* strain Y22, which our group had identified, can produce good SR [25], is a specific and valuable material for sweetpotato SR research. In this work, we chose the Y22 as material, identified its expansin gene family and analyzed their functions involved in SR formation, which could be helpful for further understanding the mechanism of SR development and functional studies of specific expansin genes.

Expansins have been characterized in many plants, but the investigations of the expansin gene family have not yet been reported in *I. trifida.* In this work, we identified 37 *ItrEXPs* in Y22 genome, which was classified into *ItrEXPA*, *ItrEXPB*, *ItrEXLA* and *ItrEXLB* subfamilies. The members of *ItrEXPA*, *ItrEXPB* and *ItrEXLB* subfamilies were fewer than those of *I. triloba*, which indicated that the gene family contracted in these subfamilies. Gene duplication may be another important driving force for species evolution [47] and cause some redundant genes. In this work, we found evidence of not only duplication of expansin genes but also triplication as well as gene loss which occurred in or after the previous WGT event [24,25]. In addition, we also found gene function redundancy occurred after gene duplication, *ItrEXPA10*, *ItrEXPA12* and *ItrEXPA15* might have evolved from one gene in the syntenic relationship (Figure 2d)*,* while only *ItrEXPA10* and *ItrEXPA15* expressed during SR development (Appendix A), which means *ItrEXPA12* may have function redundancy in the evolutionary history, that may constitute an important driving force for SR formation. The potential *cis*-elements analysis showed that *ItrEXPs* might be involved in the crosstalk between different signaling pathways to regulate the SR growth and development.

In cucumber (*Cucumis sativus* L.), the yeast two-hybrid assay demonstrated that the putative auxin transporter (numerous spine (NS)) could be interacted with CsEXLA2, which is involved in the development of numerous spines [48]. In cotton (*Gossypium hirsutum* L.), GA treated the cultured ovules correlated with enhanced expression of XTH and expansin and promoted the fiber elongation [49]. These former studies implied that expansins could be interacted with development responsive proteins, auxin-responsive proteins and GA-regulated proteins. Here, the protein interaction networks showed that ItrEXPs might interact with different type of proteins (Figure 5). Among them, the development responsive proteins might be one of the most important interaction factors, which could cleave (such as XTHs, PME5, QRT3, XTR6) and synthesize (such as CEV1s, KIN13A and part of XTHs) the cell walls (Figure 5b), and the expression of these genes in Y22 SR development exhibited up-regulation (Figure 8), therefore we deduced that ItrEXPs might combine with cell wall broken related proteins to degrade the pectin and glycan and separate the cellulose microfibrils, then the biosynthesis related proteins participated in the cell wall extension and reconstruction [2,39], thereby cell expansion and SR enlargement. This characterized analysis could be enrich our understanding about expansin gene family in *I. trifida*.

Expansins are cell wall-related proteins and play important roles in plant growth and development. In *A. thaliana*, overexpression of soybean *GmEXLB1* was shown to promote the growth of lateral roots [50], *AtEXPA7*, *AtEXPA17*, *AtEXPA18* and *AtEXLA2* were also reported to be involved in the root growth and development [6,7], and *OsEXPA8*, *OsEXPA10* and *OsEXPB2* were related to the root hair growth and root elongation in rice [8,9,10]. However, there is little research on expansins in SR development. In this work, we found, during SR development in the Y22, 22 of 37 expansin gene family members expressed (Figure 6b), *ItrEXPAs* and *ItrEXLBs* were down-regulated, and *ItrEXPBs* were up-regulated while *ItrEXLAs* not obviously changed, indicating that the expansin gene family might play an important role in SR development, and different subfamilies might have different functions. Notably, *ItrEXPs* showed tissue-specific expression in YSR, and the main roles of *ItrEXPAs* and *ItrEXPBs* might be in the cambium, while the *ItrEXLA02* and *ItrEXLBs* in the stele. The anatomic structures of Y22 SRs in our former work showed that the cell proliferation was surrounding the vessels and then the cell expansion along with starch accumulation, while the meristem was located nearby the vessels in the stele (Figure 4 of the reference) [25], therefore the higher expressions of *ItrEXLBs* in stele might be good at SR enlargement.

AR can develop into pencil root, fibrous root or SR, and AR developing into ISR is the key stage for SR formation and continuous enlargement. The SR was formed by secondary thickening of AR formed on the belowground nodes of transplanted stem cuttings [18,25]. In Y22, compare with ISR stages, the expression levels of *ItrEXLB05*, *ItrEXLB07* and *ItrEXLB08* in the AR stage were more than ten times higher (Figure 6b,c), indicating that the higher expression of these genes in the AR stage may play a key role for AR swelling to ISR, and this laid a foundation for continuous swelling and SR formation. The pencil root has thicker cortex and fibrotic small stele [51], while *ItrEXLBs* were more expressed in stele than in cambium, which might be beneficial to prevent stele fibrosis and promote cell expansion and starch accumulation, contributing to SR enlargement. Besides, we found ItrEXLA02 and sweetpotato IbEXPL1 were highly homologous, and clustered together in the phylogenetic tree (Appendix A). *IbEXPL1* has been reported to be related to sweetpotato SR development [16]. The expression level of *ItrEXLA02* remained high during Y22 SR development (Figure 6b) and had a similar expression pattern in each part of the YSRs of Y22 and sweetpotato (Figure 7f). All the results indicated that *ItrEXLA02* might be involved in SR growth and development. Taken together, our research enriched our understanding the functions of *EXLA* and *EXLB* subfamilies genes, especially for their roles in SR development, which is valuable for further research.

Above all, the systematic analysis of our work provides new insights into the structure, evolution, and predicted function of the expansin gene family members in *I. trifida*, could promote the study of specific expansin genes involved in SR development, and could be helpful for further use in sweetpotato genetic enhancement.

## Figures and Tables

**Figure 1 genes-13-01043-f001:**
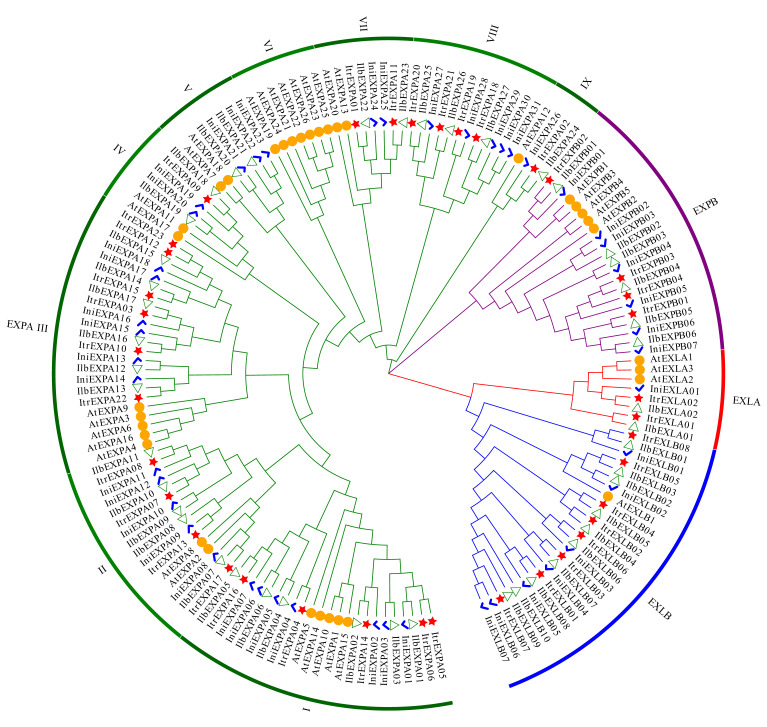
Phylogenetic tree of expansin proteins. The phylogenetic tree was constructed by using the sequences of all expansin proteins in Y22 (Itr: 37), *I. nil* (Ini: 46) (Appendix A), *I. triloba* (Ilb: 45) (Appendix A) and *A. thaliana* (AT: 35). The genes of the same species are highlighted by the same marker and color. Red stars: Y22, Blue checkmarks: *I. nil*, Green empty triangles: *I. triloba*, Orange circles: *A. thaliana*. The subclass labels I to IX of EXPA are alternately marked with dark green and green.

**Figure 2 genes-13-01043-f002:**
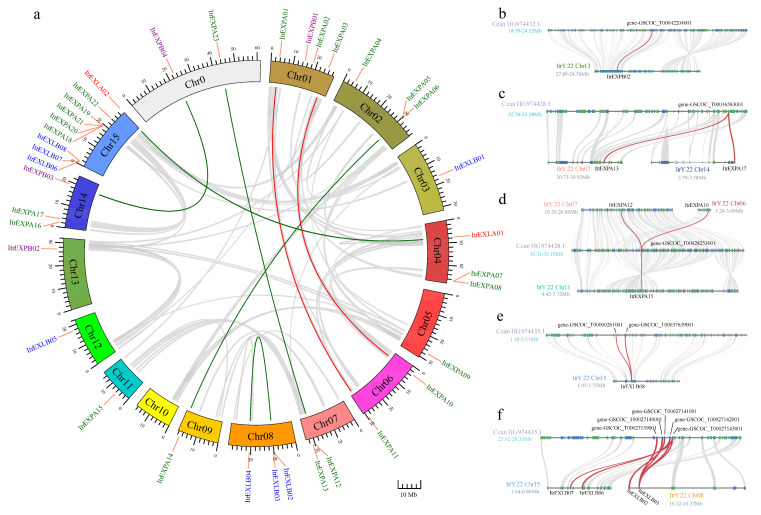
Chromosomal distribution and syntenic relationships of Y22 *ItrEXPs*. (**a**) The syntenic relationships of *ItrEXPs* in Y22 genome. The Chr0 means the scaffold that is unanchored to the chromosome level. The gray lines show the syntenic regions. The genes located in syntenic regions are marked with red lines; otherwise, they are marked with green lines. (**b**–**f**) Microsynteny analysis of expansin genes between the Y22 and *C. canephora* genomes. The expansin genes and their orthologous syntenic genes in *C. canephora* are linked by red lines, and the others are linked by gray lines. One-to-one, one-to-two, one-to-three, two-to-one and four-to-one syntenic genes between *C. canephora* and Y22 genomes are shown in (**b**), (**c**), (**d**), (**e**), and (**f**), respectively.

**Figure 3 genes-13-01043-f003:**
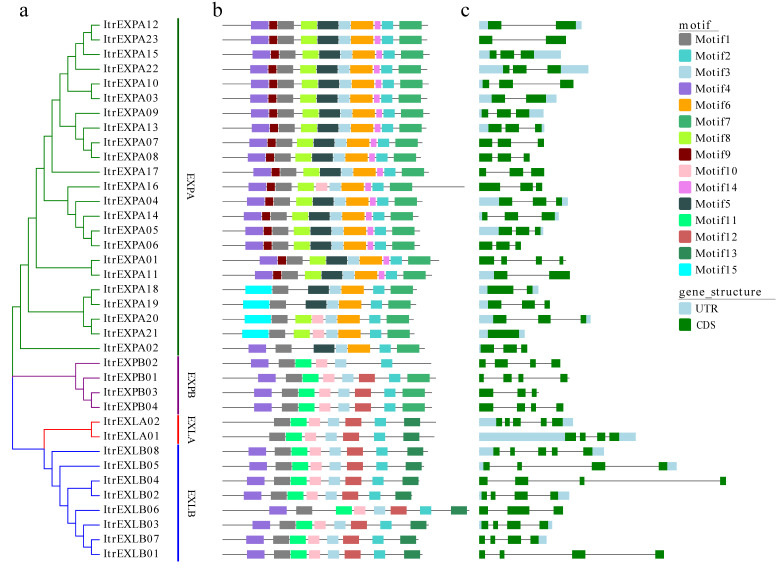
Phylogenetic relationships, motif compositions and gene structure of expansins in Y22. (**a**) Phylogenetic tree of 37 ItrEXPs in Y22. The expansins were classified into α-expansin (ItrEXPA), β-expansin (ItrEXPB), expansin-like A (ItrEXLA) and expansin-like B (ItrEXPB) groups. (**b**) Different motif compositions of ItrEXPs; the conserved motifs are represented by boxes with different colors, and their protein sequences are listed in Appendix A. (**c**) Gene structure organization of *ItrEXPs*; the introns and exons are also marked in the figure.

**Figure 4 genes-13-01043-f004:**
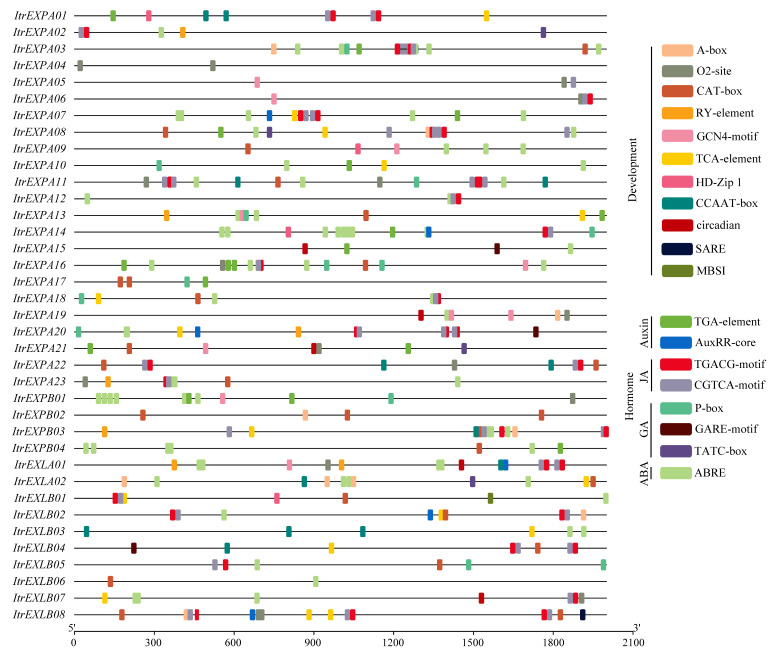
Potential *cis*-element prediction in the promoter regions of *ItrEXPs*. Left part shows the locations of *cis*-elements in the promoter regions, right part shows the *cis*-element types and their symbols.

**Figure 5 genes-13-01043-f005:**
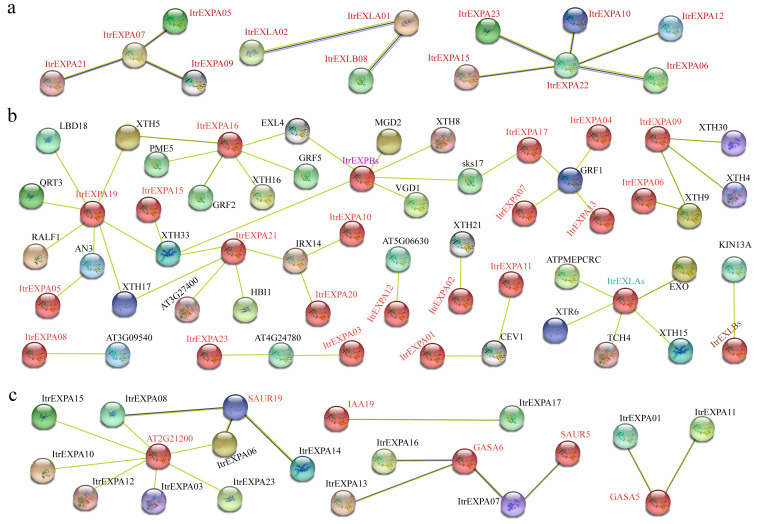
The potential interaction network of ItrEXPs. (**a**) The potential interaction network between ItrEXPs. (**b**) The potential interaction network between ItrEXPs and development related proteins. ATPMEPCRC, putative pectinesterase/pectinesterase inhibitor 26; AN3, arabidopsis grf1-interacting factor 1; CEV1, cellulose synthase a catalytic subunit 3; GRF5, growth-regulating factor; HBI1, basic helix-loop-helix (bhlh) DNA-binding superfamily protein; IRX14, nucleotide-diphospho-sugar transferases superfamily protein; LBD18, LOB domain-containing protein 18; RALF1, rapid alkalinization factor 1; sks17, SKU5 similar 17; XTH, xyloglucan endotransglucosylase/hydrolase protein; AT3G09540, AT3G27400, AT4G24780, PME5, and QRT3, pectin lyase-like superfamily protein; XTR6, probable xyloglucan endotransglucosylase/hydrolase protein 23; TCH4, xyloglucan endotransglucosylase/hydrolase family protein; EXO, phosphate-responsive 1 family protein; EXL4, exordium like 4; MGD2, monogalactosyldiacylglycerol synthase 2; VGD1, plant invertase/pectin methylesterase inhibitor superfamily; KIN13A, P-loop containing nucleoside triphosphate hydrolases superfamily protein. (**c**) The potential interaction network between ItrEXPs and hormone related proteins. AT2G21200, SAUR15, and SAUR19 belong to SAUR-like auxin-responsive protein family; GASA5, GA-regulated protein 5; GASA6, GA-regulated family protein; IAA19, IAA inducible 1. Lines between proteins represent protein-protein interaction.

**Figure 6 genes-13-01043-f006:**
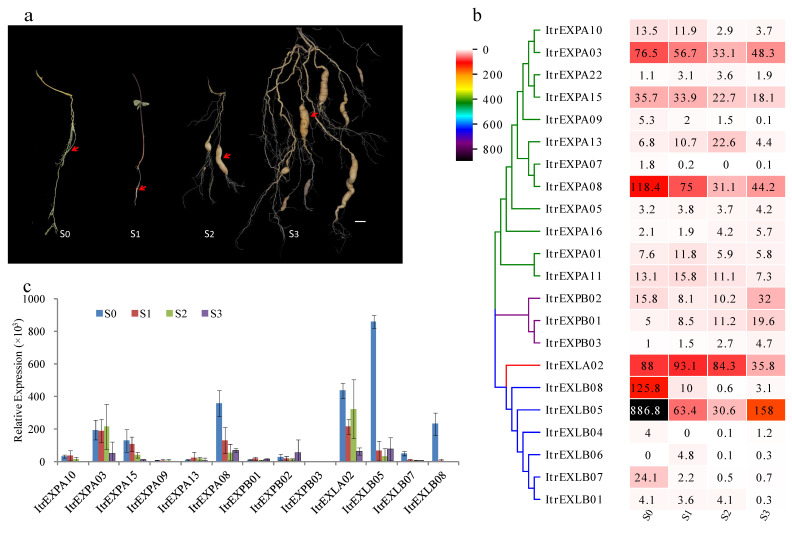
Expression profiling of *ItrEXPs* during the SR development of Y22. (**a**) Four typical stages (red arrows) of Y22 SR development. S0, AR stage; S1, ISR stage; S2, YSR stage; S3, MSR stage. Bar = 2 cm. (**b**) Expression patterns of *ItrEXPs* expressed during SR development. The expression value (RPKM) of expansin genes greater than 1 are showed in the heatmap. (**c**) qRT–PCR results of the randomly selected *ItrEXPs*.

**Figure 7 genes-13-01043-f007:**
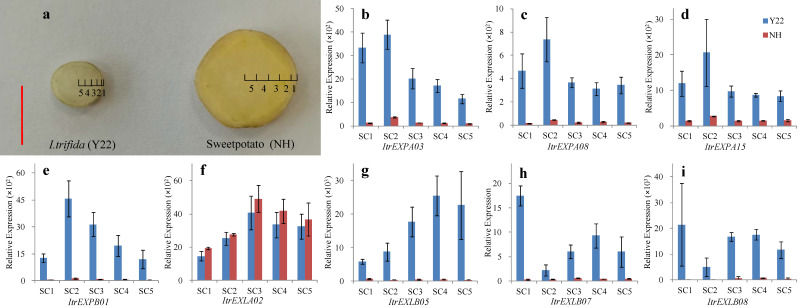
The expression of eight *ItrEXPs* in SR. (**a**) The YSRs of *I. trifida* (Y22) and sweetpotato cv. Nancy Hall (NH). Red bar, 1 cm. The number represents the SC1, SC2, SC3, SC4 and SC5, respectively. SC1: the outer section of the cortex; SC2: the inner part of the cortex and outermost part of the stele; SC3: the outer part of the stele; SC4: the middle part of the stele; SC5: the inner part of the stele. (**b**–**i**) The relative expression of *ItrEXPA03*, *ItrEXPA08*, *ItrEXPA15*, *ItrEXPB01*, *ItrEXLA02*, *ItrEXLB05*, *ItrEXLB07* and *ItrEXLB08* in the SRs of Y22 and NH, respectively.

**Figure 8 genes-13-01043-f008:**
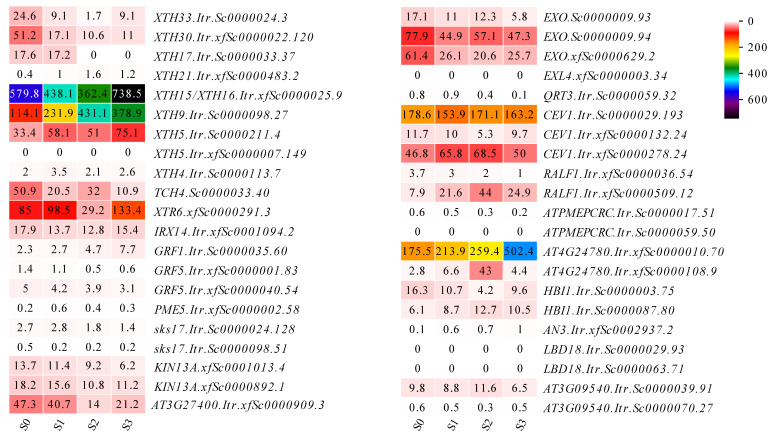
Gene expression of development related proteins that interacted with expansins in Y22 SR development. The gene ID of *I. trifida* (Y22) added later at protein gene name was the protein gene name of *I. trifida* (Y22).

**Table 1 genes-13-01043-t001:** Details of the 37 *ItrEXPs* in Y22 genome.

Subfamily	Gene Name	Gene ID	pI	Mw/kD	Length/AA	Subcellular Localization
EXPA	*ItrEXPA01*	Itr.Sc0000011.164	8.58	29.74	273	cell wall
EXPA	*ItrEXPA02*	Itr.xfSc0000576.12	10.42	28.18	254	cell wall
EXPA	*ItrEXPA03*	Itr.Sc0000007.290	9.48	27.92	257	cell wall
EXPA	*ItrEXPA04*	Itr.Sc0000051.50	9.00	26.88	251	cell wall
EXPA	*ItrEXPA05*	Itr.Sc0000046.34	9.34	26.05	248	cell wall
EXPA	*ItrEXPA06*	Itr.Sc0000046.36	9.34	26.05	248	cell wall
EXPA	*ItrEXPA07*	Itr.xfSc0000007.14	8.07	26.96	251	cell wall
EXPA	*ItrEXPA08*	Itr.xfSc0000007.15	8.07	26.54	249	cell wall
EXPA	*ItrEXPA09*	Itr.Sc0000078.10	9.38	28.29	260	cell wall
EXPA	*ItrEXPA10*	Itr.Sc0000001.78	9.60	27.98	259	cell wall
EXPA	*ItrEXPA11*	Itr.xfSc0000049.46	8.56	28.66	263	cell wall
EXPA	*ItrEXPA12*	Itr.Sc0000003.235	9.28	27.58	258	cell wall
EXPA	*ItrEXPA13*	Itr.Sc0000003.134	8.79	27.55	256	cell wall
EXPA	*ItrEXPA14*	Itr.Sc0000025.53	8.81	26.02	246	cell wall
EXPA	*ItrEXPA15*	Itr.Sc0000014.125	9.27	28.29	260	cell wall
EXPA	*ItrEXPA16*	Itr.Sc0000087.3	6.28	31.91	305	cell wall
EXPA	*ItrEXPA17*	Itr.Sc0000013.190	7.05	28.18	259	cell wall
EXPA	*ItrEXPA18*	Itr.xpSc0079065.36	6.87	26.43	244	cell wall
EXPA	*ItrEXPA19*	Itr.xpSc0079065.38	6.11	26.42	243	cell wall
EXPA	*ItrEXPA20*	Itr.xpSc0079065.39	9.35	26.50	240	cell wall
EXPA	*ItrEXPA21*	Itr.xpSc0079065.40	6.00	26.29	241	cell wall
EXPA	*ItrEXPA22*	Itr.xfSc0000000.33	9.82	27.89	257	cell wall
EXPA	*ItrEXPA23*	Itr.xfSc0001203.1	9.30	27.68	258	cell wall
EXPB	*ItrEXPB01*	Itr.xfSc0000185.16	6.29	28.52	269	cell wall
EXPB	*ItrEXPB02*	Itr.Sc0000024.30	8.74	28.41	262	cell wall
EXPB	*ItrEXPB03*	Itr.xfSc0000047.33	6.19	28.09	263	cell wall
EXPB	*ItrEXPB04*	Itr.xfSc0001448.1	6.92	28.04	263	cell wall
EXLA	*ItrEXLA01*	Itr.Sc0000050.70	5.13	28.57	266	cell wall
EXLA	*ItrEXLA02*	Itr.Sc0000020.14	6.96	29.41	268	cell wall
EXLB	*ItrEXLB01*	Itr.xpSc0079072.22	7.99	27.39	251	cell wall
EXLB	*ItrEXLB02*	Itr.xfSc0000061.14	5.71	25.49	238	cell wall
EXLB	*ItrEXLB03*	Itr.xfSc0000061.13	6.73	28.99	259	cell wall
EXLB	*ItrEXLB04*	Itr.xfSc0000239.16	8.96	26.56	248	cell wall
EXLB	*ItrEXLB05*	Itr.Sc0000071.6	6.14	28.24	253	cell wall
EXLB	*ItrEXLB06*	Itr.xfSc0000002.83.1	6.59	32.70	310	cell wall
EXLB	*ItrEXLB07*	Itr.xfSc0000002.86	8.73	27.01	246	cell wall
EXLB	*ItrEXLB08*	Itr.xfSc0000002.103	4.62	27.46	258	cell wall

## Data Availability

The RNA-seq datasets accession number of the database of National Center for Biotechnology Information (NCBI) is SUB11069337, under the BioProject number: PRJNA362521. The plant material could be available from the author M. Li.

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
