# Peer review of "Genome-Wide Identification and Expression Analysis of Expansin Gene Family in the Storage Root Development of Diploid Wild Sweetpotato Ipomoea trifida"

_genes, 2022, doi:10.3390/genes13061043_

Round 1
Reviewer 1 Report
The authors provided good quality work because their claims are supported by a well-conducted study and high-quality findings.
Author Response
Thank you.
Reviewer 2 Report
Authors evaluated the role of expansin gene family in storage root development of diploid, wild Ipomoea trifida, a model plant related to edible sweetpotato. Research is well planned and performed, results generally support presented conclusions. The English style and grammar should be checked in the manuscript. Minor improvements should be introduced.
Rewrite following sentences or improve typographic errors (localization in lines): 46-49, 51, line 66-combing?, 66-67
Paragraph 2.6- how the RNA was isolated, how the quality of RNA was assessed?, concentration and volume of cDNA libraries, how the poor quality sequences were removed- applied software, equipment used to RNAseq.
Paragraph 2.7- how the RNA was isolated, how the quality of RNA was assessed?, how the remnants of genomic DNA were removed?, conditions of RT and PCR cycling reaction, citation of previous used of reference gene or analysis using Bestkeeper or related tool, citation of Livak and Schmittgen method.
Authors could write if interactions presented in section 3.6 were confirmed experimentally (Y2H, CoIP) in other studies, if so add several sentences describing it in the Discussion section.
Author Response
Dear reviewer,
Thanks for your useful comments and suggestions. We adopted and modified the manuscript appropriately, and all the corrections were marked in the manuscript.
Sincerely,
Meifang Peng
